# PatchGame: Learning to Signal Mid-level Patches in Referential Games

**Kamal Gupta**
kampta@umd.edu

**Gowthami Somepalli**
gowthami@umd.edu

**Anubhav Gupta**
anubhav@umd.edu

**Vinoj Jayasundara**
vinoj@umd.edu

**Matthias Zwicker**
zwicker@umd.edu

**Abhinav Shrivastava**
abhinav@cs.umd.edu

University of Maryland, College Park

## Abstract

We study a referential game (a type of signaling game) where two agents communicate with each other via a discrete bottleneck to achieve a common goal. In our referential game, the goal of the speaker is to compose a message or a symbolic representation of "important" image patches, while the task for the listener is to match the speaker's message to a different view of the same image. We show that it is indeed possible for the two agents to develop a communication protocol without explicit or implicit supervision. We further investigate the developed protocol and show the applications in speeding up recent Vision Transformers by using only important patches, and as pre-training for downstream recognition tasks (e.g., classification).

## 1 Introduction

The ability to communicate using language is a signature characteristic of intelligence [60]. Language provides a structured platform for agents to not only collaborate with each other and accomplish certain goals, but also to represent and store information in a compressible manner. Most importantly, language allows us to build infinitely many new concepts by the composition of the known concepts. These qualities are shared by both the natural languages used in human-human communication and programming languages used in human-machine communication. The study of the evolution of language can hence give us insights into intelligent machines that can communicate [54].

Our goal in this paper is to develop and understand an emergent language, *i.e.*, a language that emerges when two neural network agents try to communicate with each other. Clark [14] argued that supervised approaches that consist of a single agent learning statistical relationships among symbols don't capture the functional relationships between the symbols *i.e.*, the use of symbols leading to an action or an outcome. Krishna et al. [43] argued the same viewpoint in the context of images. We, therefore, resort to the recent works in emergent language [5, 30, 42, 45, 46, 74] which show that a communication protocol can be developed or learned by two or more cooperative agents trying to solve a task. The choice of the task is quintessential since the language derives meaning from its use [78]. We choose a task where two agents, a speaker, and a listener, play a *referential game*, a type of signaling game first proposed by Lewis [48]. The speaker agent receives a target image and sends a message to the listener. The message consists of discrete symbols or words, capturing different parts of the image. The listener receives another view of the target image, and one or more distractor

---

Code is available at https://kampta.github.io/patch-game.

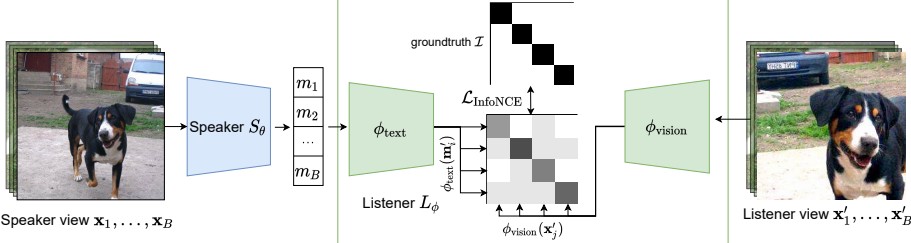

**Figure 1:** Overview of the Referential Game. We generate two random views of every image in the given batch of images. The speaker takes one of the views as the input and generates a sequence of symbols (or message). The listener takes the message sent by the speaker and the second view of the image, and projects both into an embedding space. Both the speaker and listener agents learn by minimizing the constrastive loss (see Eq. 3) between between the views in this embedding space.

images. The goal of the speaker and the listener agents is to maximize the agreement between the message and the target image. Fig. 1 illustrates the overview of the proposed referential game.

In computer vision, a number of attempts have been made to represent images as visual words [38], with a focus on low-level feature descriptors such as SIFT [49], SURF [6], *etc.*. Recent works in deep learning have attempted to describe the entire image with a fixed number of discrete symbols [58, 59, 63], however, we postulate that large images contain a lot of redundant information and a good visual representation should focus on only the "interesting" parts of the image. To discover what constitutes the interesting part of the image, we take inspiration from the works on **mid-level patches** [18, 37, 70], the patches in the image that are both *representative* and *discriminative* [28, 70]. This means they can be discovered in a large number of images (and hence representative), but simultaneously they should also be discriminative enough to set an image apart from the other images in the dataset. Hence, the speaker agent in our paper focuses on computing a symbolic representation in terms of these mid-level patches, as opposed to the entire image.

To summarize, we propose PatchGame, a referential game formulation where given an image, the speaker sends discrete signal in terms of mid-level patches, and the listener embeds these symbols to match them with another view of the same image in the presence of distractors. Compared to previous works [22, 30, 45], we make the following key changes:

- Agents in the some of the prior works [22, 30, 45] have access to a pre-trained network, such as AlexNet [44] or VGG [69], for extracting features from images. In this work, the agents rely on training on a large scale image dataset, and invariance introduced by various image augmentations, to learn the language in a self-supervised way [53].

- We propose a novel patch-based architecture for the speaker agent, which comprises of two modules: (1) PatchSymbol, a multi-layered perceptron (MLP) that operates at the patch-level and converts a given image patch into a sequence of discrete symbols, and (2) PatchRank, a ConvNet that looks at the complete image and ranks the importance of patches in a differentiable manner.

- We introduce a novel transformer-based architecture for the listener agent, consisting of two modules: (1) a language module that projects the message received from the speaker to a latent space, and (2) a vision module that projects the image into the latent space. We use a contrastive loss in this latent space to train both the speaker and the listener agents simultaneously.

- We propose new protocols to evaluate each of the speaker and listeners' modules.

We assess the success of PatchGame via qualitative and quantitative evaluations of each of the proposed component, and by demonstrating some practical applications. First, we show that the speaker's PatchRank model does indicate important patches in the image. We use the top patches indicated by this model to classify ImageNet [16] images using a pre-trained Vision Transformer [19] and show that we can retain over 60% top-1 accuracy with just half of the image patches. Second, the listener's vision model (ResNet-18) can achieve upto 30.3% Top-1 accuracy just by using k-NN ($k = 20$) classification. This outperforms other state-of-the-art unsupervised approaches [28, 63] that learn discrete representations of images by 9%. Finally, we also analyze the symbols learned by our model and the impact of choosing several hyperparameters used in our experiments.

## 2   Related Work

**Referential games.** Prior to the advent of deep learning, significant research in the field of emergent communications has shown that a communication protocol can be developed or learned by agents by playing a language game [2–5, 39, 42, 74, 77]. However, the agents employed in these works were typically located in a synthetic world and made several assumptions about the world such as the availability of disentangled representations of objects with discrete properties. More recent works [15, 23, 36, 45–47, 73, 75] have employed deep learning methods to develop a discrete language for communication between the agents. Lazaridou et al. [45] used neural network agents represented by a MLP to communicate concepts about real-world pictures. They used a fixed-sized message composed of a large vocabulary for their communication. Bouchacourt and Baroni [9], Evtimova et al. [22], Havrylov and Titov [30] relax this assumption and allow communication via variable-length sequences. Havrylov and Titov [30] allows the speaker agent to use an LSTM [33] to construct a variable-length message. Havrylov and Titov [30], Lazaridou et al. [46] show that even when we allow agents to use variable-length sequences to represent a message, they tend to utilize the maximum possible sequence to achieve the best performance (in terms of communication success).

The idea of using a Gumbel-softmax distribution [34, 51] with the straight-through trick [7] for learning a language in multi-agent environment was concurrently proposed by Mordatch and Abbeel [56] and Havrylov and Titov [30]. They show that we can achieve a more stable and faster training by using this technique as compared to reinforcement learning used in several other approaches.

**Evaluating emergent communication.** Evaluating the emergent language turns out to be an equally challenging research problem. Existing approaches use the successful completion of the task or the correlation between learned language and semantic labels as evaluation metrics. Lowe et al. [50] and Keresztury and Bruni [40] show that simple task success might not be a good or sufficient metric for evaluating the success of a game. They discuss heuristics and advocate measuring both positive signaling and positive listening independently to evaluate agents' communication. Andreas [1] provides a way of evaluating compositional structure in learned representations.

**Discrete Autoencoders.** In parallel to the works on emergent communication, there is a large body of research on learning discrete representations of images using some form of autoencoding or reconstruction [20, 28, 29, 57, 58, 62, 63, 65] without labels. The focus of VQ-VAE [58] and VQ-VAE-2 [63] is to learn a discrete bottleneck using vector quantization. Once we can represent any image with these discrete symbols, a powerful generative model such as PixelCNN [58, 66], or transformer [61, 76] is learned on top of these symbols to sample new images. PatchVAE [28] achieves the same using Gumbel-Softmax and imposes an additional structure in the bottleneck of VAEs. We argue that because of mismatch in the objectives of reconstruction and visual recognition tasks, each of these models trained using reconstruction-based losses do not capture meaningful representations in the symbols.

**Self-supervised learning in vision.** Self-supervised learning (SSL) methods, such as [10–13, 26, 32, 55, 81] have shown impressive results in recent years on downstream tasks of classification and object detection. Even though the bottleneck in these methods is continuous (and not discrete symbols), these methods have been shown to capture semantic and spatial information of the contents of the image. Unlike SSL methods, neural networks representing the two agents in our case, do not share any weights. Also, note that the continuous nature of representations learned by SSL techniques is fundamentally different from the symbolic representations used in language. And indeed, we show that a k-Nearest Neighbor classifier obtained from the continuous representations learned by SSL methods can perform better than the one obtained using Bag of Words (or symbols). However, to the best of our knowledge, our work is one of the first attempts to make representations learned in a self-supervised way more communication- or language-oriented.

**Comparison with Mihai and Hare [52, 53].** [52, 53] extends Havrylov and Titov [30] by training a speaker and listener agents end-to-end without using pre-trained networks. However, the prior works [30, 52, 52] use a **top-down** approach to generate discrete representation (or a sentence) for an image, i.e., they compute an overall continuous embedding of the image and then proceed by generating one symbol of the sentence at a time using an LSTM. The computational cost of LSTMs is prohibitive when length of a sentence is large, which is needed to describe complex images. The transformers, on the other hand, require constant time for variable length sentences at the cost of increased memory (in the listener agent). However, generating the variable length sentences with the

speaker agent using transformers is non-trivial. To solve this, we propose a **bottom-up** approach, *i.e.*, we first generate symbols for image patches and combine them to form a sentence. This approach allows for computationally efficient end-to-end training. Further, it allows the speaker to compose symbols corresponding to different parts of the image, instead of deducing it from a pooled 1D representation of the image.

## 3    PatchGame

We first introduce various notations and the referential game played by the agents in our work. We provide further details of architectures of the different neural network agents, as well as the loss function. We also highlight the important differences between this work and prior literature. Code and pre-trained models to reproduce the results are provided.

### 3.1    Referential Game Setup

Fig. 1 shows the overview of our referential game setup. Given a dataset of $N$ images $\{\mathbf{x}_i\}_{i=1}^N$, we formulate a referential game [48] played between two agents, a speaker $S_\theta$ and a listener $L_\phi$ as follows: As in the setting of Grill et al. [26], we generate two "random views" for every image. A random view is generated by taking a $224 \times 224$ crop from a randomly resized image and adding one or more of these augmentations - color jitter, horizontal flip, Gaussian blur and/or solarization. This prevents the neural networks from learning a trivial solution and encourages the emergent language to capture invariances induced by the augmentations. Given a batch of $B$ images, we refer to the two views as $\{\mathbf{x}_i\}_{i=1}^B$ and $\{\mathbf{x}_i'\}_{i=1}^B$. In each iteration during training, one set of views is presented to the speaker agent and another set of the views is shown to the listener agent.

The speaker $S_\theta$ encodes each image $\mathbf{x}_i$ independently into a variable length message $\mathbf{m}_i$. Each message $\mathbf{m}$ is represented by a sequence of one-hot encoded symbols with a maximum possible length $L$ and a fixed size vocabulary $V$. The space of all possible messages sent by the speaker is of the order $|V|^L$. The input to the listener $L_\phi$ is the batch of messages $\{\mathbf{m}_i\}_{i=1}^B$ from the speaker, and the second set of random views of the batch of images. The listener consists of a language module $\phi_{\text{text}}$ to encode messages and a vision module $\phi_{\text{vision}}$ to encode images. The goal of the listener is to match each message to its corresponding image.

Specifically, for a batch of $B$ (message, image) pairs, $S_\theta$ and $L_\phi$ are jointly trained to maximize the cosine similarities of $B$ actual pairs while minimizing the similarity of $(B^2 - B)$ incorrect pairs. For a target message $\mathbf{m}_j$, the image $\mathbf{x}_j'$ (augmented view of $\mathbf{x}_j$) acts as the target image while all the other $(B-1)$ images act as distractors. And vice versa, for the image $\mathbf{x}_k'$, the message $\mathbf{m}_k$ (encoded by the speaker $S_\theta(\mathbf{x}_k)$) acts as the target message while all the other $(B-1)$ messages act as distractors. We use the following symmetric and contrastive loss function, also sometimes referred to as InfoNCE loss in previous metric-learning works [59, 72].

$$\mathcal{L}_{\text{text}} = -\sum_{j=1}^B \log \frac{\exp(\phi_{\text{text}}(\mathbf{m}_j) \cdot \phi_{\text{vision}}(\mathbf{x}_j')/\tau)}{\sum_{k=1}^B \exp(\phi_{\text{text}}(\mathbf{m}_j) \cdot \phi_{\text{vision}}(\mathbf{x}_k')/\tau)} \tag{1}$$

$$\mathcal{L}_{\text{vision}} = -\sum_{k=1}^B \log \frac{\exp(\phi_{\text{text}}(\mathbf{m}_k) \cdot \phi_{\text{vision}}(\mathbf{x}_k')/\tau)}{\sum_{j=1}^B \exp(\phi_{\text{text}}(\mathbf{m}_j) \cdot \phi_{\text{vision}}(\mathbf{x}_k')/\tau)} \tag{2}$$

$$\mathcal{L} = (\mathcal{L}_{\text{text}} + \mathcal{L}_{\text{vision}})/2 \tag{3}$$

where $\tau$ is a constant temperature hyperparameter.

The game setting used in our work is inspired from Lazaridou et al. [45] and Havrylov and Titov [30], but there are important differences. Both our speaker $S_\theta$ and listener $L_\phi$ agents are trained from scratch. This makes the game setting more challenging, since agents cannot use the pre-trained models which have been shown to encode semantic and/or syntactic information present in natural language or images. Our training paradigm, where we show different views of the same image to the speaker and listener, is inspired by the recent success of self-supervised learning in computer vision. Empirically, we observe that this leads to a more stable training and prevents the neural networks from learning degenerate solutions. However, in contrast with such self-supervised approaches, our goal is to learn a discrete emergent language as opposed to continuous semantic representations. We discuss the differences in the architecture of the two agents in the following sections.

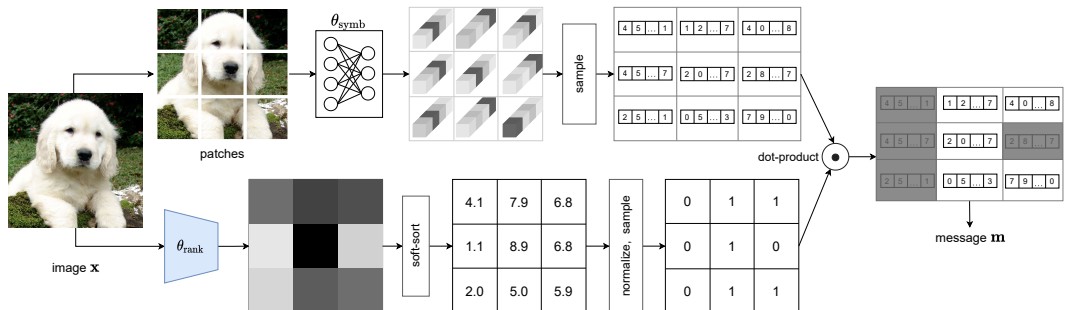

**Figure 2:** Speaker agent architecture. The upper branch represents the PatchSymbol, $\theta_{\text{symb}}$ module and the lower branch represents PatchRank, $\theta_{\text{rank}}$ module. Each of the modules take the raw image as the input. PatchSymbol computes a message for each patch of pre-defined size, using a fixed size vocabulary and message length. PatchRank uses a ConvNet to compute weights or the importance of each patch. Note that the symbols for a patch are context independent, however the importance of a patch depends on the context. The speaker agent combines the output of each of the models and sends a variable length message to the listener.

## 3.2 Speaker agent architecture

A desirable characteristic of the speaker agent is that it should be able to encode "important" components of images with a variable length sequence of discrete symbols. Previous works [9, 22, 30] have achieved this by first converting the image into a continuous deterministic latent vector and then using an LSTM network [33] to generate a sequence of hidden states, and sample from this sequence of hidden state until a special end of sequence token (or maximum length) is reached. As observed by [30, 46], in order to achieve the minimum loss, the model ends up always using the maximum allowable length. In our experiments as well, we observed that having an LSTM makes the training slow and does not achieve the objective of encoding images in variable length sequences. We propose leverage two separate modules in the speaker agent $S_\theta$ to circumvent this problem - the first module called PatchSymbol ($\theta_{\text{symb}}$) is a 2-layer MLP that computes patch-level embeddings for the image, the second module called PatchRank ($\theta_{\text{rank}}$) is a small ConvNet that computes rank or importance of each patch in the image.

**PatchSymbol, $\theta_{\text{symb}}$.** The idea of encoding an image at the patch level is inspired by the works on discovering mid-level patches [18, 37, 70] in images. We use a simple 2-hidden layer MLP, to encode each $\mathbb{R}^{C \times S^2}$ dimensional image patch $\mathbf{x}_{\text{patch}}$ to $l$ vectors of log of probabilities $[\log p_1^1, \ldots, \log p_V^1], \ldots, [\log p_1^l, \ldots, \log p_V^l]$. Here $C$ is the number of (color) channels in the input image or patch, $S$ is the spatial dimension of a square patch, $V$ is the size of the vocabulary used to represent a single symbol, and $l$ is the number of symbols used to encode each patch. Hence an image of size $\mathbb{R}^{C \times H \times W}$ can be encoded using $K = \frac{HW}{S^2}$ patches, each consisting of $l$ symbols. The vectors of log probabilities allow us to sample from a categorical distribution of $V$ categories, with a continuous relaxation by using the Gumbel-softmax trick [34, 51]. For a given vector $[\log p_1^j, \ldots, \log p_V^j]$, we draw i.i.d samples $g_i^j$ from the Gumbel$(0, 1)$ distribution [27] and get a differentiable approximation of $\arg\max$ as follows:

$$\left[[\log p_1^1, \ldots, \log p_V^1], \ldots, [\log p_1^l, \ldots, \log p_V^l]\right] = \text{MLP}(\mathbf{x}_{\text{patch}}) \tag{4}$$

$$y^j = \text{one\_hot}(\arg\max_i[\log p_i^j]) \sim \left\{ \frac{\exp((\log p_i^j + g_i^j)/\tau_s))}{\sum_{k=1}^V \exp((\log p_k^j + g_k^j)/\tau_s))} \right\}_{i=1}^V, \forall j \tag{5}$$

$$\theta_{\text{symb}}(\mathbf{x}_{\text{patch}}) = \mathbf{y} = \left\{y^j\right\}_{j=1}^l \tag{6}$$

where $\tau_s$ controls how close the approximation is to $\arg\max$. The final output of the $\theta_{\text{symb}}$ network for the entire image $\mathbf{x}$ is $L$ one-hot encoded $V-$dimensional symbols, where $L = l \times \frac{HW}{S^2}$. In all our experiments, we fix $V = 128$ and $l = 1$.

**PatchRank, $\theta_{\text{rank}}$.** An image might have a lot of redundant patches encoded using the same symbols. The goal of the $\theta_{\text{rank}}$ network is to give an importance score to each patch. Since importance of a patch depends on the context and not the patch alone, we use a small ResNet-9 [31] to compute an importance weight for each of the $K = \frac{HW}{S^2}$ patches. One possible way to use these importance weights is to simply normalize them between $(0, 1)$ and repeat the Gumbel-Softmax trick to sample

important patches. The listener network $L_\phi$ would see only the message consisting of "important" patches. However, we empirically observed that a simple min-max or L2-normalization allows the network to assign high weights to each patch and effectively send the entire sequence of length $L$ to the listener. Instead, we propose to use a *differentiable ranking* algorithm by Blondel et al. [8] to convert the importance weights to soft-ranks $\{1, \ldots, K\}$ in $O(K \log K)$ time. This method works by constructing differentiable operators as projections onto the convex hull of permutations. Once we have the vector of soft-ranks $\mathbf{r} \in \mathbb{R}^K$, we normalize the ranks and sample binary values again using a special case of the Gumbel-softmax trick for Bernoulli distributions [34, 51] as

$$\mathbf{r}(\mathbf{x}) = \arg\max_{\pi \in \Sigma} \langle \text{CNN}(\mathbf{x}), \rho_\pi \rangle \tag{7}$$

$$\theta_{\text{rank}}(\mathbf{x}) \sim \text{Bern}\left(\frac{1}{K}\mathbf{r}(\mathbf{x})\right) \tag{8}$$

where $\text{CNN}(\mathbf{x})$ are the importance weights obtained by applying a ResNet-9 to the image $\mathbf{x}$, $\Sigma$ is the set of all $K!$ permutations, and $\rho_\pi$ are the ranks corresponding to the permutation $\pi \in \Sigma$. We refer the reader to [8] for a detailed description of the soft-sort algorithm. Therefore, the final symbols encoded by the speaker agent, $S_\theta$, is given by:

$$S_\theta(\mathbf{x}) = \mathbf{m} = \theta_{\text{symb}}(\mathbf{x}) \cdot \theta_{\text{rank}}(\mathbf{x}) \tag{9}$$

### 3.3 Listener agent architecture

As discussed in §3.1, the listener agent $L_\phi$ consists of a language module $\phi_{\text{text}}$ and a vision module $\phi_{\text{vision}}$. We implement $\phi_{\text{text}}$ using a small transformer encoder [76]. We prepend a $\langle\text{CLS}\rangle$ token at the beginning of each message sequence received from the speaker [17, 29], and use the final embedding of $\langle\text{CLS}\rangle$ to compute the loss described in Eq. 3. We implement $\phi_{\text{vision}}$ using a small vision transformer [19], and follow a similar procedure as in the text module to obtain the final image embedding. Both the text and vision modules use a similar transformer encoder architecture (no weight sharing) with 192 hidden size, 12 layers and 3 attention heads. Following [11, 13, 81], we add a high dimensional projection at the end of the last layer before computing the loss function.

### 3.4 Training

Each of the weights of the speaker and listener agents $\{\theta_{\text{symb}}, \theta_{\text{rank}}, \phi_{\text{vision}}, \phi_{\text{text}}\}$ are optimized jointly during the training. We use a 2-layer MLP for $\theta_{\text{symb}}$, ResNet-9 [31] for $\theta_{\text{rank}}$, a ResNet-18 [31] for $\phi_{\text{vision}}$, and a small transformer encoder (hidden size = 192, 3 heads, 12 layers) for $\phi_{\text{text}}$. All the experiments are conducted on the training set of ImageNet [16], which has approximately 1.28 million images from 1000 classes. We create a training and validation split from the training set by leaving aside 5% of the images for validation. After obtaining the final set of hyper-parameters, we retrain on the entire training set for 100 epochs. We use Stochastic Gradient Descent (SGD) with momentum and cosine learning rate scheduling. In order to train the stochastic component of the Speaker, we use the straight-through trick [7]. We reiterate that the speaker and listener do not share weights, and the only supervision used is the InfoNCE loss defined in Eq. 3. Please refer to the appendix and code attached in the supplementary material for more details.

## 4 Experiments

We evaluate the success of communication in the referential game and impact of various hyper-parameters on the success. Also, following the work of Lowe et al. [50], we evaluate the emergent communication in two primary ways. In section 4.1, we measure *positive signaling*, which means that $S_\theta$ sends messages relevant to the observation. In section 4.2, we measure *positive listening*, which indicates that the messages are influencing the $L_\phi$ agent's behavior.

### 4.1 Positive Signaling - Visualizing Patch Ranks from $\theta_{\text{rank}}$

We first visualize the output of PatchRank module as a heatmap overlayed on the original image in Fig. 3. Most important patches are colored towards 'red' and the least important ones are colored towards 'blue'. The figure shows that our PatchRank can capture important and discriminative parts of the images. In case of images of various animals and plants, the model assigns the highest important

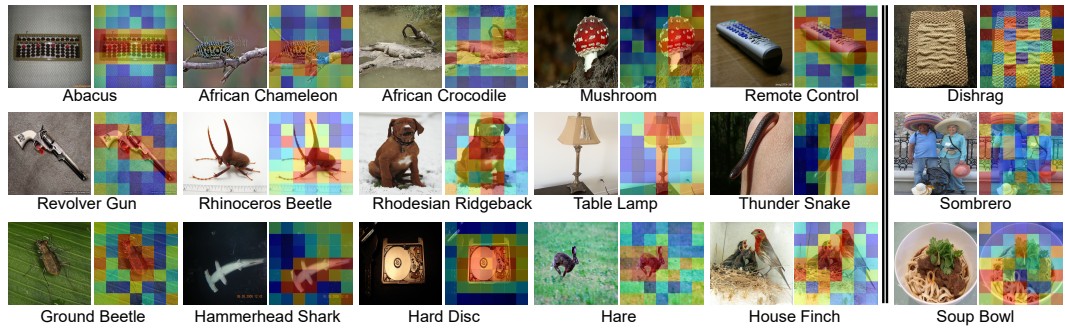

| Abacus | African Chameleon | African Crocodile | Mushroom | Remote Control | Dishrag |

| Revolver Gun | Rhinoceros Beetle | Rhodesian Ridgeback | Table Lamp | Thunder Snake | Sombrero |

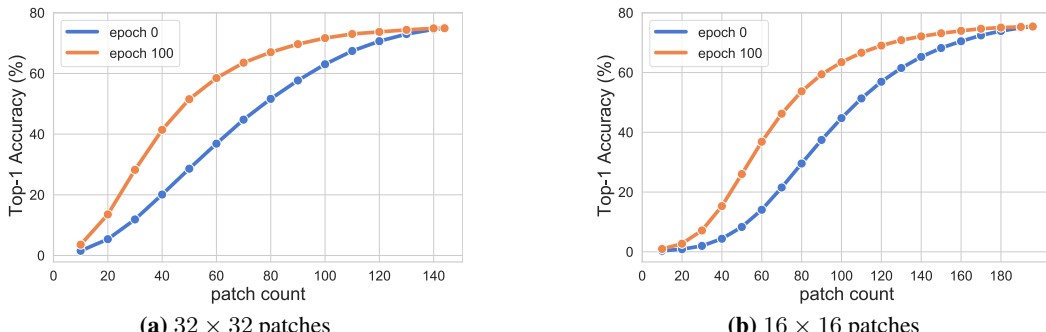

| Ground Beetle | Hammerhead Shark | Hard Disc | Hare | House Finch | Soup Bowl |

**Figure 3:** Heat maps based on Patch Importance. Red represents the most important patches and Blue the least important. Labels are listed for better understanding and are not used during training. We have used $224 \times 224$ images and $32 \times 32$ patches to get 49 non-overlapping patches per image. Model successfully separates the primary object in an image - the "device" in the "Abacus" image, "rabbit" in the "hare" image etc. *(as shown by higher concentration of yellow-to–red patches on these areas)*. Some failure cases are shown as well on the right.

**(a)** $32 \times 32$ patches

**(b)** $16 \times 16$ patches

**Figure 4:** Impact of number of patches used during evaluation in a pre-trained ViT [19]. The performance of a ViT drops if we provide fewer patches during inference. However a PatchRank model (which is a small ResNet-9) can provide important patches to ViT during inference with minimal loss in Top-1 accuracy.

to discriminative body parts. For the inanimate objects such as abacus or revolver the model is able to distinguish between the foreground and the background. Note that although approaches such as GradCam [67] can provide pixel-level importance heatmaps, they require extensive supervision. Our method on the other hand is self-supervised. We also show some of the failure cases when discriminative patches in the image cover majority of the image on the rightmost 2 columns of Fig. 3.

## 4.2 Positive Signaling - Image Classification with subset of patches provided by $\theta_{\text{rank}}$

Recently proposed Vision Transformers (ViT) by Dosovitskiy et al. [19] have gained popularity because of their simplicity and performance. These models treat an image as a sequence of $N \times N$ patches, use an MLP to convert the patch into an embedding, and finally use these set of patches to perform classification. We first note that both the inference time and memory consumption of ViT depend largely on the length of sequence, because of the $O(N^2)$ self-attention operation. Secondly, the performance of the Vision Transformers drops if instead of using all patches, we only use a subset of patches during inference. This provides us a simple way to evaluate the $\theta_{\text{rank}}$ module. We artificially constrain the number of patches available to ViT during inference. We measure the Top-1 Accuracy of ViT using only $k$ allowed patches. The selection of the patches is done by the $\theta_{\text{rank}}$ model.

We consider two different pre-trained ViT [19] models, one trained using $32 \times 32$ patches, on the images of size $384 \times 384$ (so the number of patches in an image is 144), while the second model is trained using $16 \times 16$ patches on the images of size $224 \times 224$ (196 patches per image). In Fig. 4a and 4b, we show the Top-1 accuracy obtained by the pre-trained ViT models using important patches predicted by $\theta_{\text{rank}}$ at different values of $k$. At the $0^{\text{th}}$ epoch (with random weights), the performance of ViT drops almost linearly as we lower the patch count $k$. At the $100^{\text{th}}$ epoch, at the end of the training, we observe that performance of the ViT does not drop as drastically.

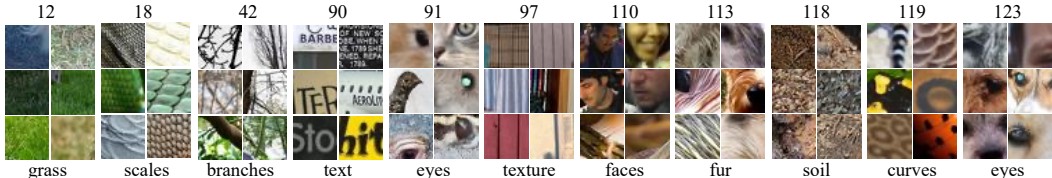

**Figure 5:** Visualizing patches corresponding to various symbols. The number in the top row corresponds to 1 of 128 vocabulary ids. 6 representative patches corresponding to each patch are shown. The bottom row corresponds to our interpretation of what concept that symbol might be capturing.

**Table 1:** Performance of Listener's Vision module $\phi_{\text{vision}}$ - Downstream classification accuracy for ImageNet dataset using k-NN (k=20)

| Method | Top-1 (%) | Top-5 (%) |
|---|---|---|
| MoCo-v2 [32] (R18) | 36.8 ($\pm$ 0.2) | 60.3 |
| VQ-VAE2 [59] | 17.2 ($\pm$ 0.6) | 30.5 |
| PatchVAE [28] (R18, $S = 16$) | 16.4 ($\pm$ 0.6) | 28.5 |
| PatchVAE [28] (R18, $S = 32$) | 21.3 ($\pm$ 0.5) | 36.2 |
| Ours (R18, $S = 16$) | 27.6 ($\pm$ 0.6) | 46.2 |
| Ours (R18, $S = 32$) | **30.3** ($\pm$ 0.5) | **49.9** |

**Table 2:** Performance of Listener's Vision module $\phi_{\text{vision}}$ - Downstream mean Average Precision for Pascal VOC

| Method | mAP-50 (%) |
|---|---|
| MoCo-v2 [32] (R18) | 65.8 |
| PatchVAE [28] ($S = 16$) | 52.2 |
| PatchVAE [28] ($S = 32$) | 54.2 |
| Ours ($S = 16$) | 58.9 |
| Ours ($S = 32$) | **61.3** |

### 4.3 Positive Signaling - Visualizing $\theta_{\text{symb}}$ symbols

As mentioned earlier, we are using a vocabulary $V = 128$ and patch size $S = 32$ in our base model. This means $\theta_{\text{symb}}$ has to map each $32 \times 32$ patch to one of the 128 available symbols. A natural way to analyze what the symbols are encoding to see is to visualize their corresponding patches. While $V = 128$ is far too small a vocabulary to describe all possible patches, we observe some interesting patterns in Fig. 5. Many of the symbols seem to adhere to specific concepts repeatedly. We observed that symbols have a lesser preference for color, but more preference for texture and shape. We discovered several symbols corresponding to textures such as grass, branches, and wood. We also noticed many symbols firing for the patches corresponding to text, eyes, and even faces. There can be multiple symbols representing single concept, *e.g.*, both symbol 91 and 123 both fire in case of eyes.

### 4.4 Positive Listening

Next, we evaluate the vision module, or $\phi_{\text{vision}}$ of the listener. We follow the protocol employed by various approaches in self-supervised learning literature. We consider the features obtained at the final pooling layer of the $\phi_{\text{vision}}$ (which is a ResNet-18 in our case). Next, we run a k-NN classification with $k = 20$ on the validation dataset. Table 1 shows the Top-1 and Top-5 % accuracy obtained on ImageNet using the listener's vision module and the baselines approaches. Although our method outperforms VQ-VAE-2 and PatchVAE (methods that learn a discrete image representation) we observe that there is still a gap between representations learned by these models as compared to the representations learned by continuous latent models such as MoCo [32]. Note that, because of the resource constraints, all results reported in the table are obtained by training ResNet-18 for only 100 epochs. The results for both MoCo-v2 and our approach continue to improve if we continue the training beyond 100 epochs (as also noted by He et al. [32]). Further, we use the listener's vision module as a pre-trained network for Pascal VOC dataset [21]. Our results are shown in the Table 2. Again, results are not competitive as compared to self-supervised counterparts such as MoCo-v2 but we outperform models with discrete bottleneck such as PatchVAE. We find that the convergence of models with discrete bottlenecks (such as this work, and PatchVAE) is slow and hence, improving the training efficiency of this class of models is an interesting future direction.

### 4.5 Ablation study

A communication iteration is successful if the receiver is able to match the message sent by the speaker to the correct target image. In our experiments, we use an effective batch size of 512 (split over 4 GPUs), so the chance accuracy of success of the speaker is $0.19\%$. We measure the Top-1 accuracy for each image of the batch and average it over validation data at the end of epoch. In Fig. 6a, we observe that too high or too low learning rates can be detrimental to the success of communication. We fix the learning rate at 0.0001 which shows the best validation performance empirically. We train

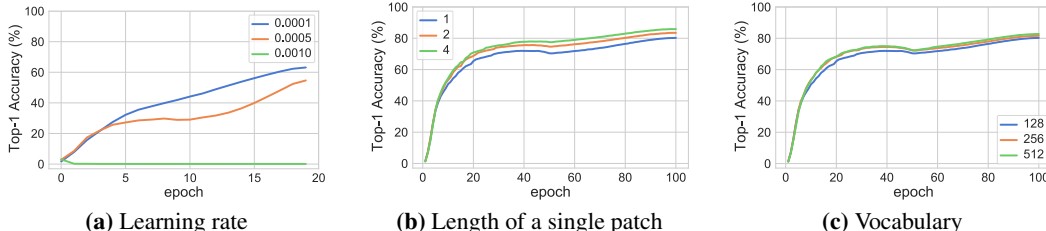

| (a) Learning rate | (b) Length of a single patch | (c) Vocabulary |

**Figure 6:** Impact of various hyperparameters on the success of communication. Top-1 accuracy denotes the percentage of messages in a batch that model was able to match to the corresponding image. Having a higher message length and vocabulary helps the model to learn faster.

our models at different vocabulary sizes, and different message lengths for 100 epochs. From Fig. 6b and 6c, we observe that having either a large vocabulary or larger message length allows the model to reach high accuracy faster. This is intuitive since the larger the space spanned by the messages is, the easier it is for receiver to distinguish between the images. A large message length increases the possible number of messages exponentially, at the cost of much larger computation cost since self-attention [17, 76] is an $O(N^2)$ operation. A large vocabulary also increases the both the span of messages and computation cost linearly.

## 5 Discussion

**Summary.** In this work, we have shown that two cooperative neural network agents can develop a variable length communication protocol to solve a given task on a large scale image dataset with only self-supervision. To the best of our knowledge, this is the first work to develop an emergent communication via mid-level patches without using any pre-trained models or external labels. We have introduced a novel mid-level-patch based architecture for the speaker agent, in order to represent only the "interesting" parts of image with discrete symbols. The listener agent learns to align the messages and images using a language and image encoders to a common embedding space. The speaker and listener agents are trained end-to-end jointly to capture invariances induced by various data augmentations in order to solve a constrastive task. We propose a number of quantitative and qualitative measures to analyse and evaluate the emerged language. We also show two major applications of the developed approach - (1) extract representative and discriminative parts of the image (2) transfer learning in image classification tasks.

**Limitations and Future Work.** There are a few limitations of our approach. Firstly, one of the applications discussed in Section 4.1 is using fewer patches for inference in ViT. Although, using fewer patches reduces the memory cost of ViT, the overhead of using another neural network for predicting the important patches means our gain is minimal. In the future, we would like to explore even faster architectures to have a bigger impact on classification speed. Second, our method only works with fixed-size square patches. Discovering arbitrary sized mid-level patches is a challenging task that we would like to address in future work. Third, the language emerged from our current model is not grounded in natural language and requires human intervention for interpretation. Going forward, we would like to ground this emerged language in the natural language.

**Broader Impact.** Like most self-supervised approaches, our approach is data hungry and trained using a large amount of unlabeled data collected from the Internet. Our model in its current form is prone to learning and amplifying the biases present in the dataset, especially if the dataset in question is not carefully curated. While the data collection and labeling has been discussed in the original paper [16], the community has focused towards imbalance and privacy violation in existing image datasets only recently [79]. A recent study [80] shows that 997 out of 1000 ImageNet categories are not 'people' categories. To compare against previous methods and allow future benchmarking, we provide results on the 2012 version of the dataset with 1.28 million images in training set and 50000 images in the validation set.

**Acknowledgements.** We thank Matt Gwilliam, Max Ehrlich, and Pulkit Kumar for reviewing early drafts of the paper and helpful comments. This project was partially funded by DARPA SAIL-ON (W911NF2020009), and Amazon Research Award to AS.

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
