# A    Training details

Both speaker and listener agents $\{\theta_{\text{symb}}, \theta_{\text{rank}}, \phi_{\text{vision}}, \phi_{\text{text}}\}$ are trained jointly during the training. All the experiments are conducted on the training set of ImageNet [16], which has approximately 1.28 million images from 1000 classes. We create a training and validation split from the training set by leaving aside 5% of the images for validation. After obtaining the final set of hyper-parameters, we retrain on the entire training set for 100 epochs. We use Stochastic Gradient Descent (SGD) with momentum and cosine learning rate scheduling. In order to train the stochastic component of the Speaker, we use the straight-through trick [7]. We use linear warmup for learning rate for 30 epochs. We also use a cosine schedule for temperature annealing for Gumbel Softmax, where we decrease the temperature from 5.0 to 1.0 in first 50 epochs and then fix the temperature to 1.0 for rest of the epochs. Please find the code and model (see the README file) attached in the supplementary material.

# B    More Ablations

**Augmentations**    We perform an additional ablation study where we train the model on ImageNet for 20 epochs with different augmentations removed and observed which ones have the most impact on downstream classification task with kNN. The results are shown in the Table 3.

**Batch Size**    In our experiments, we observed that higher batch size leads to both stable and improved downstream performance which is intuitive since it allows for contrastive loss to be more accurate. In all our experiments, we use the maximum batch size that could fit in 12GB GPU memory (128 images per GPU). The contrastive loss is computed over the aggregate batch size over 4 GPUs (and hence the effective batch size of 512 images). For larger batch sizes, we scale the learning rate linearly [25]. Table 4 below shows the downstream classification performance of the model at different batch sizes (trained for 20 epochs).

**Table 3:** Ablation study for augmentations

| Augmentation | Top-1 (%) |
|---|---|
| Baseline | 26.3 |
| Remove color jitter | 23.2 |
| Remove random resized crops | 22.9 |

**Table 4:** Ablation study for batch sizes

| Batch Size | Top-1 (%) |
|---|---|
| 128 | 47.2 |
| 256 | 49.1 |
| 512 | 49.9 |

# C    Visualizing more symbols

Fig. 7 shows some additional examples of symbol ids and their possible meaning. We observed that most symbol ids fire for consistent patterns. An interesting observation is that model doesn't focus a lot on the color but the textures and shapes in the images. One of the limitations of our approach is the fixed-size grid used in the architecture which restricts the patch size to $32 \times 32$ or $16 \times 16$. This restriction is important for the efficient training. However, it results in some symbols capturing only partial concepts such as part of a face or text. In future works, we seek to address this limitation.

# D    Topographic Similarity

We analyse the topographic similarity between the learned messages and images from the validation dataset as following: We take 10 random images from each category. Then we compute pairwise Jaccard Similarity (JS) between all $\binom{10}{2} = 45$ image pairs for that category. JS corresponds to a simple intersection over union (IoU) of the set of symbols in the messages corresponding to two images. This indicates how similar the images are with respect to message generated by the Speaker (note that even though speaker generates a sequence of symbol, we analyse it as a set for this analysis). For each pair, we also compute Learned Perceptual Image Patch Similarity (LPIPS) metric [82] using off-the-shelf VGG model. We draw a scatter plot between the two similarity metrics as show in Fig. 8 We observe that some categories, such as 'toucan', 'filing cabinet', and 'crab' have very high correlation, while there are some categories with almost no correlation between LPIPS and Jaccard similarity. The mean and median correlation across all categories is 0.25 and 0.21.

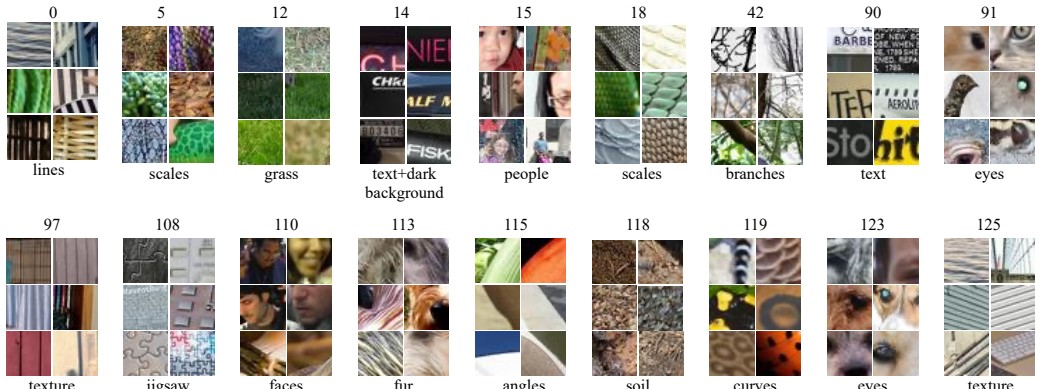

**Figure 7:** Visualizing some more symbols. The number in the top row corresponds to one of the 128 vocabulary ids (from 0-127). 6 representative patches corresponding to each patch are shown. The bottom row corresponds to our interpretation of what concept that symbol might be capturing.

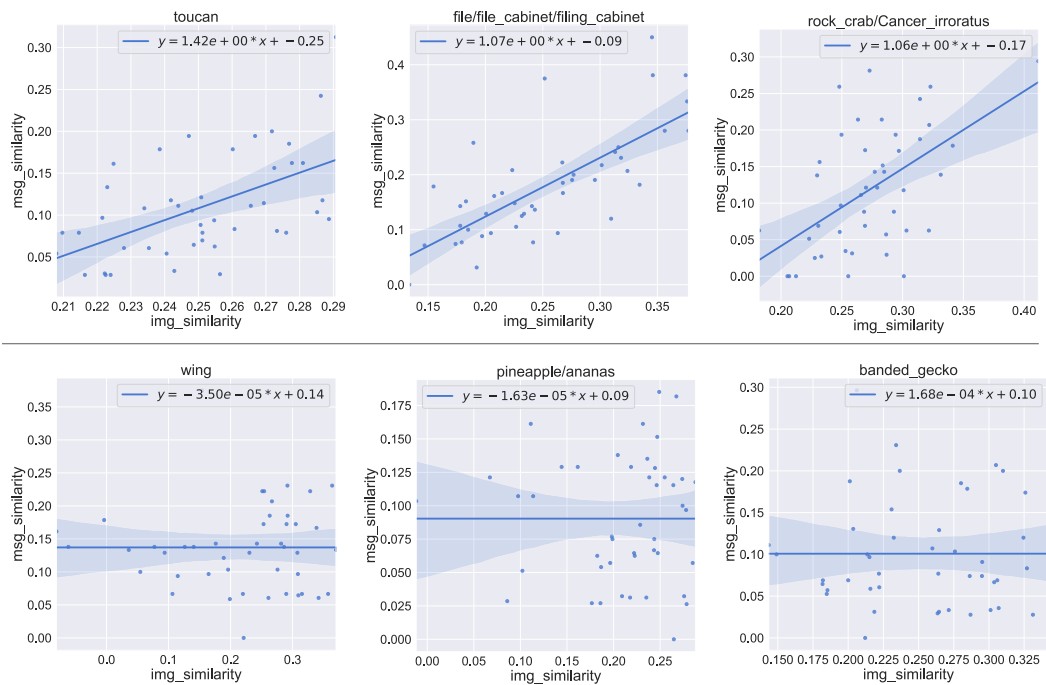

**Figure 8:** Topographic similarity computed as pairwise correlation between Jaccard similarity of messages and LPIPS perceptual similarity of images [82]. The top row shows the 3 categories with highest topographic similarity and the bottom row shows 3 categories with lowest topographic similarity. Note that the slope coefficient is in the scientific notation $a \times 10^b$

# E   Variable length messages

We analyze the number of symbols and number of unique symbols appearing in each message, as illustrated in Fig. 9. Fig. 9a shows that that all each message has at least 7 unique symbols, without any particular symbol excessively recurring within a message. No image uses more than 16 unique symbols even though maximum allowable symbols that can be used by an image is 49 (from the vocabulary of 128 symbols). Fig. 9b shows the distribution obtained when we consider all symbols (and not just unique) for a message. The distribution looks like a gaussian and this is because of the design choice we made in our architecture. Since for a given image, we sample the patch ranks from

a normalized and sorted list of ranks, only about half of the symbols, end up getting selected and hence the peak of the distribution is at 24-25 (half of 49).

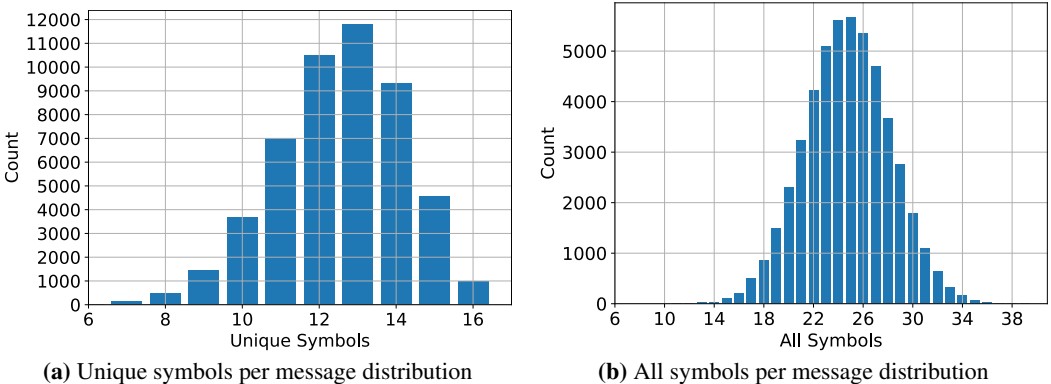

(a) Unique symbols per message distribution

(b) All symbols per message distribution

**Figure 9:** Analysis of Patch Symbols. We investigate the number of unique symbols appearing in an image (on the x-axis). y-axis shows the number of images in the validation set with the corresponding number of unique symbols. Most images use 13 unique symbols (maximum possible is 49) for representation.

## F  Relationship between length of messages and images

We try to analyze qualitatively the relationship between the images and the length of message representation as generated by the speaker. We sort all the images in the validation set by the length of the message and report some of the images with very long and very short message lengths in Fig. 10. We observe that, on an average, images with lengthier messages appear more complex visually with a lot of clutter or variety of objects in the image. Images with shorter message length usually have a single object.

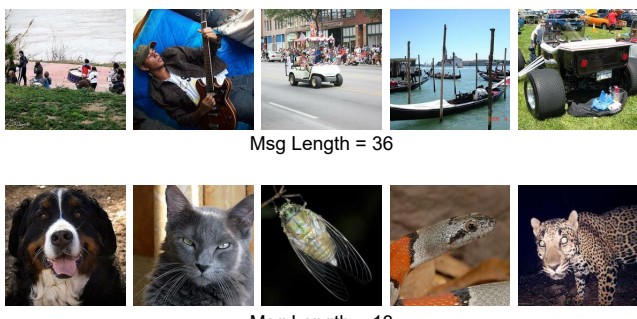

**Figure 10:** Complex images require long messages while visually simple images like the ones shown above can be represented with short messages.

## G  Visualizing saliency maps and PatchRank

We qualitatively compare the patch ranks generated by our method with the corresponding saliency maps generated using a standard technique. Since the saliency maps provide us with important regions of an image, the patches deemed as important by our method should overlap with the said salient regions. Note that the saliency method uses a supervised classification model and class label to generate the pixelwise importance heatmap. Our method on the other hand generates these plots in an unsupervised manner.

Following this hypothesis, we extract the saliency maps for the ImageNet validation set with the XGradCAM [24] approach, using the ResNet-50 [31] model with true category labels obtained from

official code repository. Fig. 11 illustrates the saliency maps and the patch ranks obtained for a few such random ImageNet validation images. We observe that important patches ranked by our model has a high correspondence with the salient regions of the images.

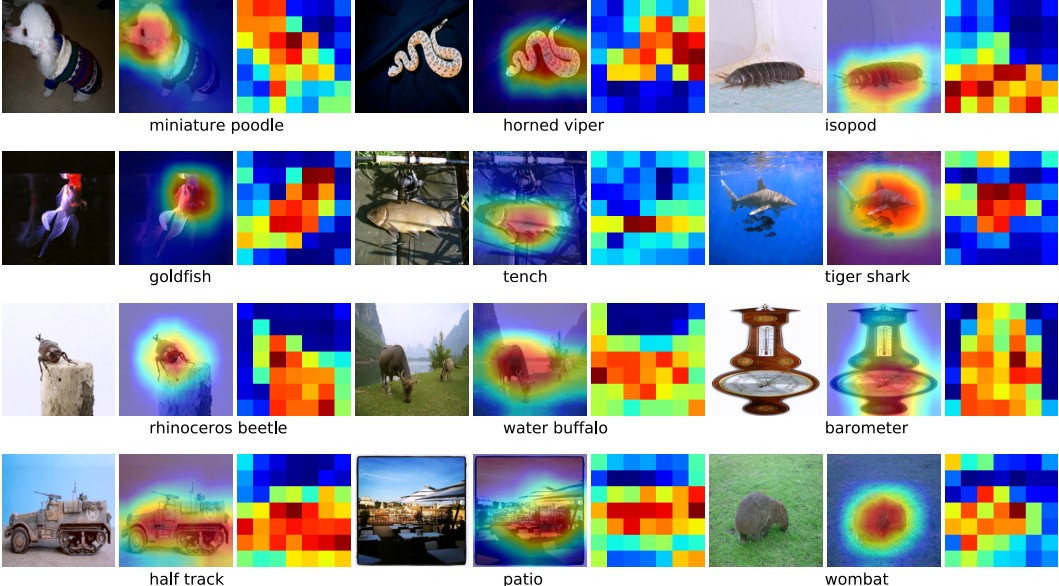

**Figure 11:** Comparison of saliency maps and the patch rank heatmaps based on importance for ImageNet validation images. Red represents the most important patches and Blue the least important. For each set of three images, the first, second and third sub-images correspond to the original image, saliency maps, and patch heatmaps respectively. Note that the saliency method uses a supervised classification model and class label to generate the pixelwise importance heatmap. Our method on the other hand generates these plots in an unsupervised manner.

## H    Analyzing the symbol distribution

We investigate the distribution of the patch symbols generated with our method. For each image in the ImageNet validation set, we compute the list of symbols generated by $\theta_{\text{symb}}$. Then we calculate the frequency that each symbol appear across the images in the validation set, as illustrated in Fig. 12. Even though the distribution is not uniform, we observe that all symbols recur often, suggesting that our generated symbols are not redundant. This complements our observation in Section 4.3 (Fig. 5 of the manuscript) where we observed that there are some symbols that correspond some very common textures and patterns such as grass or lines, while some symbols on the other hand capture more specific concepts such as faces, or eyes.

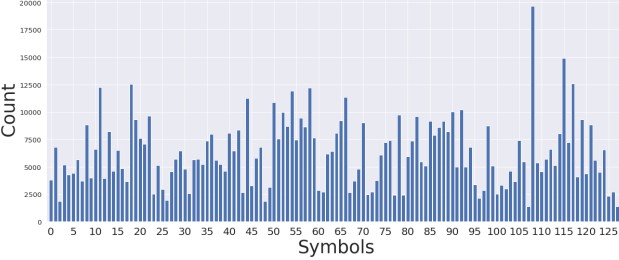

**Figure 12:** Analysis of Patch Symbols. We investigate the distribution and characteristics of symbols generated by our method for the ImageNet validation set. x-axis shows symbol ids (between 0 and 127), while y-axis shows the number of times that symbol appeared in the representation of images in validation set. Symbols have non-uniform rate of appearance in the images. However, most of the symbols are utilized by the model.

# I   Positive Signaling - Visual Bag of Words Classification with $\theta_{\text{symb}}$

Representing the image as visual bag of words is a well-known technique in image retrieval works [71]. Since our task is to represent image with words as well, we devise the following way to evaluate the PatchSymbol module or $\theta_{\text{symb}}$. For each image in the ImageNet training and validation set, we compute the message or the list of symbols generated by $\theta_{\text{symb}}$ (and filtered according to $\theta_{\text{rank}}$). We then compute a feature for each image using the weighted frequency of symbols in the image or *tf-idf* [35]. We then train a simple complement naive bayes classifier [64] on top of these feature vectors. For benchmarking, we choose two prior works that encode images to symbols and have shown strong results in generative modeling. **VQ-VAE-2** or Vector Quantized Variational AutoEncoder was initially proposed by Oord et al. [58] and later improved by Razavi et al. [63] for both conditional and unconditional large scale image generation. We retrain the model for 100 epochs at $256 \times 256$ resolution used in the paper using [68]'s repository. **PatchVAE** by Gupta et al. [28] proposes a structured variant of Variational AutoEncoders [41]. We use the authors's code and retrain their models for a $7 \times 7$ bottleneck (corresponding to $S = 32$ patch size), and a $14 \times 14$ bottleneck (corresponding to $S = 16$ patch size), each for 100 epochs.

Table 5 shows the Top-1 and Top-5 % classification accuracies on ImageNet. The patch symbols generated with our method outperform those generated by VQ-VAE-2 and PatchVAE in the downstream classification task by factors of 45% and 25% respectively. Note that classification accuracy is far below the ones that can be obtained using standard techniques, however the goal of this evaluation is to demonstrate that the symbols do capture meaningful information.

**Table 5:** Performance of Speaker's PatchSymbol module $\theta_{\text{symb}}$ - Downstream classification accuracy for ImageNet dataset using visual BoW method with Complement Naive Bayes

| Method | Top-1 (%) | Top-5(%) |
|---|---|---|
| VQ-VAE2 [58] | 0.88 ($\pm$ 0.03) | 5.02 |
| PatchVAE [28] ($S = 32$) | 1.02 ($\pm$ 0.07) | 4.48 |
| PatchVAE [28] ($S = 16$) | 1.00 ($\pm$ 0.08) | 5.04 |
| Ours ($S = 32$) | **1.28** ($\pm$ 0.06) | **6.10** |
| Ours ($S = 16$) | 1.20 ($\pm$ 0.06) | 5.94 |