# OpenReview forum: "PatchGame: Learning to Signal Mid-level Patches in Referential Games"
_NeurIPS.cc/2021/Conference — NeurIPS 2021 Poster_

### Official Review · Reviewer_2BvU · 2021-07-12

**Rating:** 6
**Confidence:** 4

**Summary:**

This paper introduces a referential game in which a speaker “sees” an image and sends a discrete message of variable length to a listener. The message is designed to represent important patches in the input image. In addition to the speaker’s message, the listener is given a batch of images, one of which is an augmented view of the speaker’s input. The listener’s goal is to find this augmented view of the input image among all images in the batch. The speaker and listener are trained with self-supervision using InfoNCE loss. The learned communication protocol is evaluated on the underlying task as well as used as a pre-training step for the downstream task of classification.

**Limitations And Societal Impact:**

Yes, the limitations and societal impact (from the perspective of training data) are discussed in the paper. The authors might also consider mentioning any potential negative applications of the proposed model.

**Main Review:**

### Strengths

This work presents a novel referential game inspired by recent developments in self-supervised learning applied to computer vision in which two views of the same input image are used to learn a representation useful for downstream applications such as classification. To the best of my knowledge, the combination of a patch-based architecture of the speaker and a transformer architecture of the listener in addition to a discrete image representation is novel and well motivated. The authors discuss the computational complexity of the proposed model. The paper is written and organized clearly. It explores an interesting research direction at the intersection of self-supervised learning and referential games and favorable classification performance when compared to existing methods such as VQ-VAE-2 and PatchVAE.

### Room for improvement
While the authors do note that resource constraints have prevented them from reporting error bars in their experiments, I believe the work would benefit from including these to support the statistical significance of the results. For example, the plots in Fig 6 are difficult to interpret because there are no error bars present and all curves are very close to one another. Also, Fig 2a does not display the curves past epoch 20.

### Questions
1. Have you explored what effect the batch size has on classification performance?
2. Just to clarify, is the x-axis of Fig 6b “Length of each patch” the number of symbols used per patch?
3. What was the motivation behind setting $l $ (the number of symbols per patch) to 1? Is it the computational cost?
4. What is meant by “meaningful representations” in line 107? Using autoencoders as a pre-training step has been shown useful for downstream tasks such as classification.

### Nitpicks
- Fig 2’s caption: “ConvNet to compute weights...”
- Line 140: use either “encode” or “embed”, not both
- Line 239: “We also show some of the failure...”
- Line 247: “the performance of the Vision Transformers drops if...”
- Line 302: “an effective batch size of 512”
- Line 332: “means our gain is minimal”
- Line 339: “from the Internet”

### During discussion
- After reading the authors' response to my comments and questions, I have decided to increase my score.

**Time Spent Reviewing:**

6

---

> ### Author Response · Authors · 2021-08-10
> **Adding standard error; ablation study on batch size and other clarifications**
>
> Thank you for your time and thoughtful feedback. We are pleased that you find the research direction interesting, our “patch based speaker architecture, transformer based listener architecture”, and “discrete image representation” novel and well-motivated, and our writing clear and organized.
>
> We address the specific concerns below -
>
> **Regarding standard error for plots and tables** - Thank you for highlighting this. For the benefit of the readers, we will include standard error on multiple runs in all tables and figures. We run each experiment with different seeds 3 times and report the mean$\pm$std error below. Specifically for figure 6, the numbers are as below. (For standard errors on Table 1 and 2, see our response to Reviewer 5nrF)
>
> | Hyperparameter/Epoch   | 1 | 5 | 10 | 15 | 20 |
> |:--- |:---:|:---:|:---:|:---:|:---:|
> | Learning Rate = 0.0001 | 1.4$\pm$0.1 | 31.5$\pm$1.4 | 47.7$\pm$1.6 | 57.4$\pm$1.2 | 64.0$\pm$0.5 |
> | Learning Rate = 0.0005 | 2.8$\pm$0.1 | 25.7$\pm$1.6 | 28.9$\pm$1.0 | 36.4$\pm$1.1 | 54.7$\pm$0.7 |
> | Learning Rate = 0.0010 | 3.1$\pm$0.1 | 0.1 | 0.1 | 0.1 | 0.1 |
> | Length of patch = 1 | 1.4$\pm$0.1 | 31.5$\pm$1.5 | 47.7$\pm$1.6 | 57.4$\pm$1.3 | 64.0$\pm$0.5 |
> | Length of patch = 2 | 1.4$\pm$0.1 | 34.2$\pm$1.4 | 53.2$\pm$1.3 | 62.1$\pm$0.9 | 68.7$\pm$0.6 |
> | Length of patch = 4 | 1.3$\pm$0.1 | 33.8$\pm$1.4 | 54.4$\pm$1.3 | 64.6$\pm$1.0 | 70.9$\pm$0.6 |
> | Vocabulary = 128    | 1.4$\pm$0.1 | 31.5$\pm$1.5 | 47.7$\pm$1.6 | 57.4$\pm$1.3 | 64.0$\pm$0.5 |
> | Vocabulary = 256    | 1.4$\pm$0.1 | 34.2$\pm$1.2 | 53.6$\pm$1.3 | 62.8$\pm$0.7 | 68.2$\pm$0.6 |
> | Vocabulary = 512    | 1.3$\pm$0.1 | 32.3$\pm$1.3 | 53.1$\pm$1.0 | 63.5$\pm$0.9 | 68.4$\pm$0.5 |
>
> **Regarding Fig. 6a** - (We assume you meant figure 6a and not 2a, if that’s not the case, kindly let us know) For most ablation studies, we train our model only till 20 epochs due to limited computational budget. The main results are reported using the model trained for 100 epochs. If the reviewer deems it necessary, in the final version, we will include the numbers till 100 epochs.
>
> We observe that the model keeps on improving in terms of the success of communication protocol, as well as downstream classification as we keep training for more epochs. This is consistent with the behavior on larger datasets such as ImageNet as reported in self-supervised learning literature. For example, MoCov2 [1] shows that performance keeps improving even at 1000 epochs.
>
> ---
>
> **Q1. Batch Size** - In our experiments, we observed that higher batch size leads to both stable and improved downstream performance which is intuitive since it allows for contrastive loss to be more accurate. In all our experiments, we use the maximum batch size that could fit in 12GB GPU memory (128 images per GPU). The contrastive loss is computed over the aggregate batch size over 4 GPUs (and hence the effective batch size of 512 images) (ref. L387-L397 of `patch_game/builder.py`). For larger batch sizes, we scale the learning rate linearly [2]. The table below shows the downstream classification performance of the model at different batch sizes (trained for 20 epochs).
>
>
> | batch size   | top-1 acc |
> |:---:|:---:|
> | 128 | 47.2 |
> | 256 | 49.1 |
> | 512 | 49.9 |
>
> ---
>
> **Q2, Q3. Symbols per patch** - x-axis in Figure 6b refers to the training epochs, but the different curves indeed correspond to the “number of symbols used to represent a single patch”. The motivation behind using multiple symbols per patch is to allow our model to represent complex patches with more than one symbol. However, increasing the number of symbols used for one patch increases the sequence length used to represent the entire image. This is not desirable since the memory cost of a single forward pass of the transformer increases quadratically with the sequence length. From Figure 6b, even though there is a slight performance gain from using larger values of $l$, we resort to using only 1 symbol per patch, since it allows us to train larger batch sizes (and hence results in faster training). Note that each image is represented by either $14\times14$ or $7\times7$ maximum possible patches (or symbols), which arguably is a large enough length to meaningfully describe an image. We will explain this further. Thank you for highlighting this.
>
> ---
>
> **Q4. Learning Meaningful Representations** - While VAEs and $\beta$-VAEs have been shown to capture some meaningful global image-level semantics (for example, changing one of the latent variables can change the pose in digits or faces), the disentanglement in the latent space of autoencoders without labels is very challenging if not impossible [6]. We believe this is because the pixel-level reconstruction loss used in auto-encoders is not very effective for recognition tasks such as classification of object detection [3, 4, 5]. The motivation behind working with mid-level patches is to fill this gap. We will clarify this in the manuscript.
>
> ---
>
> **Potential Negative Applications** - We believe we are still in the early phases of understanding the behavior of emergent languages. Like most deep learning models, our model in its current form is prone to learning and amplifying the biases present in the dataset (especially if the dataset in question is not carefully curated). For example, compression and image retrieval are two potential applications of representing images in discrete space. In each of the applications, we need to carefully analyze how the model is behaving in the class-imbalance regimes. We will include this in our discussion.
>
> ---
>
> We hope we addressed your concerns and kindly urge you to consider increasing your score.
>
> ---
>
> **References**
>
> 1. Chen, Xinlei, et al. "Improved baselines with momentum contrastive learning." arXiv preprint arXiv:2003.04297, 2020.
> 2. Goyal, Priya, et al. "Accurate, large minibatch sgd: Training imagenet in 1 hour." arXiv preprint arXiv:1706.02677, 2017.
> 3. Higgins, Irina, et al. "$\beta$-VAE: Learning basic visual concepts with a constrained variational framework." ICLR, 2016.
> 4. Donahue, Jeff, et al. "Adversarial feature learning." ICLR, 2017.
> 5. Gupta, Kamal, et al. "Patchvae: Learning local latent codes for recognition." CVPR, 2020.
> 6. Locatello, Francesco, et al. "Challenging common assumptions in the unsupervised learning of disentangled representations." international conference on machine learning. PMLR, 2019.

---

> > ### Comment · Reviewer_2BvU · 2021-08-22
> > **Increasing my score**
> >
> > Thank you for the detailed response and for addressing my concerns! I have decided to increase my score.

---

> > > ### Author Response · Authors · 2021-08-23
> > > **Thanks**
> > >
> > > Thank you for increasing the score. Your feedback has indeed been very helpful to us in improving the paper.

---

### Official Review · Reviewer_Hpoa · 2021-07-14

**Rating:** 6
**Confidence:** 5

**Summary:**

This paper presents a visual referential game played by a pair of agents. Compared to existing work in this space, the game is a variant of those studied previously as the communication is based directly on encoded patches (rather than a latent vector describing the whole image), and the agent models diverge from prior work in their architectural construction.


**Limitations And Societal Impact:**

Limitations are covered adequately. I am not convinced that the current Broader Impact section in the paper adds anything relevant to the discussion - my suggestion would be to rework this into something much more tangible about how people might benefit (or not) from potential future machinery that benefits from this work (for example, how an emergent communication system between agents might be used, or how variable-length discrete compressed representations could be utilised).


**Main Review:**

Originality
-----------

Unfortunately, one of the main claims to originality the paper makes is that prior work in emergence of language using visual referential games all looks at models with pretrained networks for visual feature extraction rather than models in which this inductive bias is removed. There was a paper at the NeurIPS 2019 Emergent Communication workshop that explored exactly this aspect and started to draw out the links between self-supervised learning (with augmentations) and emergent communication:

    Avoiding hashing and encouraging visual semantics in referential emergent language games. Daniela Mihai and Jonathon Hare. 3rd NeurIPS Workshop on Emergent Communication. 2019. https://arxiv.org/abs/1911.05546

A quick search on arxiv suggests the same authors might also have followed this up (https://arxiv.org/abs/2101.10253) with a longer paper.

With this in mind, there are still some bits of the proposed model which are potentially interesting - particularly the game setup and the agent architecture. Overall, I think the proposed setup is an interesting variation of what has been done before, although I'd have liked to have seen stronger links made to prior work incorporating visual attention --- for example attending to different parts of an image to generate discrete messages is in essence what the Show, Attend & Tell model (https://arxiv.org/pdf/1502.03044.pdf) does (albeit in a fully supervised setting). I believe the approach to generating variable length message sequences is novel, although I do have questions (below).


Quality
-------

Technically the proposed model and accompanying quantitive evaluation results appear generally sound. As a reader I would have appreciated more to be said about the variance of the model from differing initializations (particularly in the experiments in section 4.6) --- it is clear that the model is very sensitive to learning rate, so one also wonders how much variance from random initializations there is in e.g. Fig 6b and 6c where the model performance under differing vocabulary and message lengths is _very_ close (is it so close that it's within the margin of error?).

I do wonder if the authors could further justify their choices of metrics for measuring performance of the model (or perhaps clarify their intent in creating the model); more specifically, in the context of emergent communication it seems strange that one would consider a bag-of-words model or a truncated-message model as a good proxy for comparing positive signaling as they do not capture the interesting parts of language that one might wish to observe the emergence of (such as evidence of compositionally, etc). Perhaps the aim was to just explore models with discrete latent spaces (and perhaps an implicit 'importance' ordering on tokens)? If so, such a criticism is obviously mute, but in such a case would it then not make sense to focus more attention on exactly what makes the proposed model perform better than e.g. VQ-VAE or PatchVAE?

Regarding PatchRank, it's not actually clear to me that the proposed approach does really control that message length much. Surely in expectation the length of every message is just ~K/2 as (approximately because of the soft-rank) half of the values in r(x) would be >K/2? Besides the potential for some small information leakage from the relaxation in the soft-rank algorithm I can't see that the model really has much ability to learn appropriate message lengths. Do you have any quantitive evidence (e.g. mean message length and its variance over many games) that the model really is actually learning variable length better (e.g. with higher variability, not necessarily shorter than the maximum) than previous models?


Clarity
-------

Overall I found the text to be easy to follow, however I think many of the points I've raised above hint to ways to improve the content and clarity.

### Minor points
- \theta_{rank} in Eqn. 8 is inconsistent with the rest of the text (and Fig. 2)


Significance
------------

Commenting on potential significance is always challenging; in this case it's not immediately obvious to me that there is that much new given the points raised above, however this could change if my queries were addressed.


Rationale for rating
----------------------
Overall my opinion is that there is little novel in the paper in its current form, although there are some potentially interesting aspects which I've tried to highlight in my review. My score reflects this. I've tried to give constructive feedback around some of the more original parts of the paper and sincerely hope these will help the authors make improvements.


**Time Spent Reviewing:**

5

---

> ### Author Response · Authors · 2021-08-10
> **Adding related work, standard error, and other clarifications**
>
> Thank you for your time and valuable feedback. We are pleased that you find the “approach to generating variable length message sequences” novel, “game setup and the agent architecture” interesting, and the quantitative evaluations sound. The specific comments are addressed below:
>
> **Comparison with Mihai et al. [1,2]** - We thank the reviewer for pointing out very important works by Mihai et al. [1,2] which extend [3] by training the network end-to-end. This is indeed an oversight on our part and we will include it in both the related work section, as well as the baselines, and will update the contributions in the introduction accordingly. That said, we would like to highlight the most important difference between our method and theirs. Each of the prior works [1,2,3] uses a **top-down** approach to generate discrete representation (or a sentence) for an image, i.e., they compute an overall continuous embedding of the image and then proceed by generating one symbol of the sentence at a time using an LSTM. For large and complex datasets, this approach is computationally inefficient since *as the length of a sentence increases, the time consumed by a single forward and backward step increases linearly in LSTM*. The transformers alleviate this problem by using constant time for variable length sentences at the cost of increased memory. While the use of transformers is straight-forward in the listener agent (assuming the complete sentences are available to the listener), *generating the variable length sentences with the speaker agent using transformers is non-trivial*. This is because a vanilla transformer generator (or decoder) uses “teacher forcing” (or known ground-truth sequences) as input to the model during training. In our case, we don’t have a ground-truth discrete representation of the image, which necessitates the use of the **bottom-up** approach as described in our paper, i.e., we first generate symbols for parts (or patches) of the image and combine them to form a sentence. This approach has *three inherent advantages* - (i) The entire game can be trained end-to-end with standard backpropagation in a computationally efficient manner, and hence, easily extensible on large datasets (ii) The patch-based approach allows the network to easily compose symbols corresponding to different parts of the image, instead of deducing it from a pooled 1D representation of the image, and (iii) arguably PatchRank allows more control on the distribution of the length of sentences (in our work, we used a ranking based regularization which ensures the expected length of the sentences is half of the maximum allowable length as we show in Section C and D).
>
> We believe both the two top-down and bottom-up approaches are complementary and important directions in developing and understanding emergent communication. Exploring the differences in the behavior of the language that emerges from these two protocols is another interesting direction for future work.
>
> Moreover, kindly note that [1,2] report the results on $32\times32$ CIFAR-10 dataset, while in our patch based approach, we use patches of size $32\times32$ for $224\times224$ images in ImageNet. Hence a direct and fair comparison of the two approaches is not straight-forward. The official code for [1,2,3] is not available and we are currently trying to replicate their results on ImageNet to provide a fair comparison. Despite our best efforts, we haven’t been able to train [1,2] on ImageNet with random initialization (using the libraries [6, 7]). We have also requested the authors for the code used in their paper and we will continue the effort for the final manuscript, either using the provided implementation, or our own implementation. We have also attached our code in the supplementary material and are committed to making our code public for the community.
>
> ---
>
> **Comparison with Xu et al. [4]** - Xu et al.’s paper is indeed a relevant work in the direction of interpretable caption generation of images with labeled data. However, the assumption of labeled training data for image captions, makes the two problem settings quite different. Even though the listener agent in our case is using self-attention to encode the symbols, it is not attending to the image directly, but to other symbols which are computed for a patch. We have attempted to understand what these symbols (which are learned in an unsupervised setting) might mean in Sec 4.3 (Fig. 5) and Sec A (Fig. 1). However, computing a pixel level attention map like [4] is a non-trivial problem and we will try to include it in the final version.
>
> ---
>
> **Regarding standard error for plots and tables** - Thank you for highlighting this. For the benefit of the readers, we will include standard error on multiple runs in all tables and figures. We run each experiment with different seeds 3 times and report the mean$\pm$std below. Specifically for Figure 6, the numbers are as below.
>
> | Hyperparameter/Epoch   | 1 | 5 | 10 | 15 | 20 |
> |:--- |:---:|:---:|:---:|:---:|:---:|
> | Learning Rate = 0.0001 | 1.4$\pm$0.1 | 31.5$\pm$1.4 | 47.7$\pm$1.6 | 57.4$\pm$1.2 | 64.0$\pm$0.5 |
> | Learning Rate = 0.0005 | 2.8$\pm$0.1 | 25.7$\pm$1.6 | 28.9$\pm$1.0 | 36.4$\pm$1.1 | 54.7$\pm$0.7 |
> | Learning Rate = 0.0010 | 3.1$\pm$0.1 | 0.1 | 0.1 | 0.1 | 0.1 |
> | Length of patch = 1 | 1.4$\pm$0.1 | 31.5$\pm$1.5 | 47.7$\pm$1.6 | 57.4$\pm$1.3 | 64.0$\pm$0.5 |
> | Length of patch = 2 | 1.4$\pm$0.1 | 34.2$\pm$1.4 | 53.2$\pm$1.3 | 62.1$\pm$0.9 | 68.7$\pm$0.6 |
> | Length of patch = 4 | 1.3$\pm$0.1 | 33.8$\pm$1.4 | 54.4$\pm$1.3 | 64.6$\pm$1.0 | 70.9$\pm$0.6 |
> | Vocabulary = 128    | 1.4$\pm$0.1 | 31.5$\pm$1.5 | 47.7$\pm$1.6 | 57.4$\pm$1.3 | 64.0$\pm$0.5 |
> | Vocabulary = 256    | 1.4$\pm$0.1 | 34.2$\pm$1.2 | 53.6$\pm$1.3 | 62.8$\pm$0.7 | 68.2$\pm$0.6 |
> | Vocabulary = 512    | 1.3$\pm$0.1 | 32.3$\pm$1.3 | 53.1$\pm$1.0 | 63.5$\pm$0.9 | 68.4$\pm$0.5 |
> ---
> ** Using Bag of Words for evaluation ** - Bag of words in language and Visual bag of words in vision have been commonly used in retrieval and classification literature, especially in many works that pre-date deep learning. One of the potential applications of finding a discrete representation of the images is that it is compact and can be very efficient to store and search by indexing. We include them in our work since we believe they can provide an interesting way to evaluate and improve discrete representation learning in the future. Our current hypothesis on why this model works better than VQ-VAE or PatchVAE is the use of contrastive loss in training instead of the reconstruction loss (L99-108).
>
> ---
>
> **Compositionality** - Note that the use of bottom-up approach makes this model compositional by design in the 2D space, however achieving and evaluating compositionality like the one that exists for natural languages, in absence of any ground truth, is an important but orthogonal research question that we leave for the future work.
>
> ---
>
> **Message Length** - Indeed the expected length of the messages is K/2 as we demonstrate in Section C and D of the appendix (also Fig. 3 and 4 of the appendix). However, we argue that this can be viewed as one of the strengths of our approach. In contrast to [3, 5], where the speaker tends to utilize the maximum possible sentence length in order to compose the message for achieving maximum possible accuracy, having a prior on the message length allows us to control the distribution of the message lengths. We agree with the reviewer that it is an important discussion point and we will move it to the main manuscript.
>
> **Broader Impact** - We believe it is very important to discuss privacy and data collection while training any large-scale models using Internet data and is very relevant to the discussion. Most deep learning models face the danger of learning and amplifying the biases present in the dataset. At the same time, we agree with the reviewer that it is important to discuss how other applications might benefit from the emergent communication. Some of the potential applications of the method could be in image retrieval or fast/secure communication between machine learning models. We will elaborate on them further in the manuscript.
>
> ---
>
> **References**
>
> 1. Mihai, Daniela, et al. "Avoiding hashing and encouraging visual semantics in referential emergent language games." NeurIPS Workshop on Emergent Communication. 2019.
> 2. Mihai, Daniela, et al. "The emergence of visual semantics through communication games." arXiv preprint arXiv:2101.10253. 2021.
> 3. Havrylov, Serhii, et al. "Emergence of language with multi-agent games: Learning to communicate with sequences of symbols." NeurIPS 2017.
> 4. Xu, Kelvin, et al. "Show, attend and tell: Neural image caption generation with visual attention." ICML 2015.
> 5. Lazaridou, Angeliki, et al. "Emergence of linguistic communication from referential games with symbolic and pixel input." ICLR 2018.
> 6. https://github.com/facebookresearch/EGG
> 7. https://github.com/Near32/ReferentialGym

---

> > ### Comment · Reviewer_Hpoa · 2021-08-22
> > **Revised score**
> >
> > Thank you for the detailed response. If the paper was modified in the way described I would be happy to increase my score. I'm not particularly fussed about having an experimental comparison with [1] or [2] although it would be a nice addition if possible; the key thing would be to situate the contributions of this paper in the context other related work.

---

> > > ### Author Response · Authors · 2021-08-23
> > > **Authors' response**
> > >
> > > Thank you! While revisions are not accepted during the review process (https://neurips.cc/Conferences/2021/PaperInformation/NeurIPS-FAQ), as we mentioned in our response, [1,2] are indeed important prior works. We will update our contributions in the context of these works, as well as include [1,2] in related work and add them as baselines if we are able to reproduce the results on ImageNet.
> > >
> > > Kindly consider increasing your score if we have addressed your concerns.

---

### Official Review · Reviewer_i9eZ · 2021-07-17

**Rating:** 7
**Confidence:** 4

**Summary:**

The authors present PatchGame, "a referential game formulation where given an image, the speaker sends discrete symbolic signals in terms of mid-level patches, and the listener embeds these symbols to match them with another view of the same image in the presence of distractors."

**Ethical Concerns:**

None identified.

**Limitations And Societal Impact:**

The broader impact section refers to "A recent study [75] shows that 997 out of 1000 ImageNet categories are not ‘people’ categories."
This sentence reads out of context, could you clarify the purpose of stating this here?

**Main Review:**

The novelty of the work is high, and as claimed it is the first work to develop an emergent communication via mid-level patches without using any pre-trained models or external labels. While I enjoyed reading the draft (the presentation is clear), I have a few questions and also comments:

1) From reading the game setting, and its root in emergent communication formation, I have the feeling that the speaker and listener model is essentially trained to encode in the speaker view image into a minimum set of patch-based representations that the listener could re-identify the image similar to the input image from a set of images. Instead of reconstructing the same image back in an encoder-decoder architecture, the game here is trying to encode the input image into a sequence of patches (with length constraint) to help another agent to identify from a set.  Is it essentially a special case of an encoder-decoder system? If not, why?

2) Have you considered and/or observed the speaker and listener learned to bypass the "important" image patch but learned to utilize artifacts (maybe a logo in the input image, and also present in the target image from the set of candidate images) to encode the message? According to the game setting, the optimized message would be the artifact, then the model won't be able to learn to rank "important" image patches right? In other words, since it is an unsupervised game setting, how to prevent the agents from "cheating" to succeed?

3) It is a comment. Have you considered utilizing this game setting to formulate a novel evaluation mechanism for visual/representation learning, rather than the debatable downstream task classification accuracy-based ones? I feel this might be the more exciting usage of your proposed game.

Other places need clarification:

Line 332 means we our gain is minimal. -> delete "we"

**Time Spent Reviewing:**

2 hours

---

> ### Author Response · Authors · 2021-08-10
> **Adding an augmentation ablation study; writing improvements**
>
> Thank you for your time and thoughtful feedback. We are happy that you enjoyed reading the draft and encouraged that you find our work highly novel with clear presentation. We address the specific comments below:
>
> **Is it an encoder-decoder system?** - Your intuition is correct; our approach can indeed be thought as a special case of an encoder-decoder system, where instead of encoding an image in a continuous space, we encode the image in a discrete space (akin to PatchVAE, VQ-VAE). However, there is a key difference to note. The goal of the decoder (or listener) is not to reconstruct the encoding but to contrast the encoding from encodings of other images. Hence, the loss is computed using multiple images, instead of vanilla per-image loss used in auto-encoder or encoder-decoder systems.
>
> ---
>
> **How to prevent the agents from cheating to succeed?** - That is an important question and one of the ways we (and several other works in self-supervised learning) prevent it is by presenting different augmentations of the same image to the speaker and listener. This way, even if the speaker agent encodes an artifact into the symbols, the listener agent’s vision module might not see the same artifact. There is still a trade-off on how much augmentations one can add for the model to be able to learn meaningful representations, and answer to that question would depend on the task and dataset at hand. We perform a small ablation study where we train the model on ImageNet for 20 epochs with different augmentations removed and observed which ones have the most impact on downstream classification task with kNN.
>
> | Augmentation  | top-1 accuracy |
> |:---|:---:|
> | Baseline                    | 26.3 |
> | remove color jitter         | 23.2 |
> | remove random resized crops | 22.9 |
>
> That said, an extensive analysis of how the sender and receiver can cheat to achieve successful communication is indeed an important direction for future work.
>
> ---
>
> **Evaluation mechanisms** - Thank you for the suggestion. One of the ways we utilized our approach to evaluate visual representation learning is to use PatchRank for randomly selecting Patches for Vision Transformer (Section 4.2). There can indeed be other exciting ways to evaluate representation learning using Referential Games which we will continue to explore in the future.
>
> ---
>
> **Language Improvements** - We will proof-read the entire paper. We will add more context in the broader impact section. Thank you for the suggestions.

---

### Official Review · Reviewer_5nrF · 2021-07-20

**Rating:** 7
**Confidence:** 4

**Summary:**

The current work proposes to study a signaling game between two agents that uses image patches instead of the entire image. Within this framework, experiments show that the agents select important patches from the image and the emergent language captures semantic information about the patches. Further, the resultant agents are used as self-supervised image classifiers and compared against existing patch-based/self-supervisory methods to show superior performance.

**Limitations And Societal Impact:**

Authors discuss the limitations and broader impact of their work in the manuscript.


**Main Review:**

**Strengths**
(S1) Leveraging mid-level semantic patches for referential games and using the resultant agent for downstream image classification task (without pretrained features) is a novel direction.

(S2) The manuscript is well-written and is easy to read and understand. In particular, it states the hypothesis clearly and places it in the context of prior work.

(S3) The authors do a good job with the experiments supporting the claims in the earlier part of the manuscript. The qualitative examples of heat maps and semantic grouping of emergent language throw more light on the proposed idea, and are useful.

**Weaknesses**
(W1) The standard deviation for performances is missing for Table 1 and 2. Given that some of these numbers are close, it is not possible to deduce if the improvements are statistically significant.

**Comments**
(C1) How generalizable are the features/images if images beyond ImageNet are used, for example MSCOCO or OpenImages?

(C2) L239-240: Is there a way to quantify how much overlap exists between important patches and the dominant objects for images in the dataset? For example, using a subset of images and computing overlap with segmentation masks for either PASCAL or other datasets with segmentation annotations?

(C3) Would a pair of agents participating in a referential game (but without patches) trained from scratch converge? If yes, what would their downstream performance on image classification be?

**Typos**
L239: shows -> show


**Time Spent Reviewing:**

2

---

> ### Author Response · Authors · 2021-08-10
> **Adding standard error; preliminary results on Pascal VOC**
>
> Thank you for your time and detailed feedback. We are pleased that you find our approach of leveraging mid-level patches for referential games novel, our manuscript well-written, and the experiments useful and well-motivated. We address the specific comments below:
>
> **[W1] Standard error for the model performance** - We thank you for pointing it out, and agree that having standard error for the performance will help readers understand the numbers better. Hence, we re-run each experiment 3 times, and state the mean$\pm$std error  below. We will also update the tables and figures in the manuscript accordingly.
>
> Performance of Speaker’s PatchSymbol module $\theta_{\text{symb}}$ (Table 1):
>
> | Method   | Top-1 (%) |
> |:--- |:---:|
> | VQ-VAE2           | 0.9$\pm$0.03 |
> | PatchVAE (S=32)   | 1.0$\pm$0.07 |
> | PatchVAE (S=16)   | 1.0$\pm$0.08 |
> | Ours (R18, S = 32)| 1.3$\pm$0.06 |
> | Ours (R18, S = 16)| 1.2$\pm$0.06 |
>
> Performance of Listener’s vision module $\phi_{\text{vision}}$ (Table 2):
>
> | Method   | Top-1 (%) |
> |:--- |:---:|
> | MoCo-v2           | 36.8$\pm$0.2 |
> | VQ-VAE2           | 17.3$\pm$0.6 |
> | PatchVAE (S=32)   | 21.5$\pm$0.5 |
> | PatchVAE (S=16)   | 16.5$\pm$0.6 |
> | Ours (R18, S = 32)| 30.3$\pm$0.5 |
> | Ours (R18, S = 16)| 27.9$\pm$0.6 |
>
>
> ---
>
> **[C1, C2] Beyond ImageNet** - Using the learned discrete symbols for a new dataset and/or a new task is a very interesting idea, but beyond the scope of current work. We did try using the listener’s vision module as a pre-trained network for Pascal VOC. Some preliminary results are shown in the table below. The results are not competitive as compared to self-supervised counterparts such as MoCo-v2 but we outperform models with discrete bottleneck such as PatchVAE. We find that the convergence of models with discrete bottlenecks (such as this work, and PatchVAE) is slow and hence, improving the training efficiency of this class of models is an interesting future direction. Regarding the overlap between segmentation masks and patches for dominant patches, for a vocabulary size of 128, and 20 VOC classes, there are a large number of permutations to consider and we will try to carefully design a quantitative metric for this purpose.
>
> | Method   | AP50 |
> |:--- |:---:|
> | MoCo-v2           | 65.8 |
> | PatchVAE (S=32)   | 54.2 |
> | PatchVAE (S=16)   | 52.2 |
> | Ours (R18, S = 32)| 61.3 |
> | Ours (R18, S = 16)| 58.9 |
>
> ---
>
> **[C3] Ablation study (no mid-level patches)**
> As pointed out by *Reviewer Hpoa* - it is indeed possible to train a model from scratch without using mid-level patches. Although it is non-trivial to use a transformer in that case, and using LSTMs makes the training computationally inefficient for large datasets such as ImageNet.
> (Also, kindly see our detailed response to the reviewer Hpoa).

---

> > ### Comment · Reviewer_5nrF · 2021-08-24
> > **Response to the author rebuttal**
> >
> > Thanks to the authors for rebuttal.
> >
> > My concerns regarding statistical significance and performance beyond ImageNet were addressed satisfactorily.
> >
> > I believe the current work has enough novelty to merit a publication at the venue, and will stick to my rating of "Accept".

---

### Author Response · Authors · 2021-08-10
**Summary of proposed changes**

Dear reviewers and AC:

We thank all of you for your time and valuable suggestions. To summarize, we propose the following changes in the manuscript

1. Add the mean and standard error to Table 1, 2 and Figure 4, 6
2. Add preliminary results on Pascal VOC
3. Add an extra ablation study on augmentations and batch sizes
4. Add ablation study for Mihai et al. and discuss it in the related work
5. Move the discussion on sentence length from the appendix to the main paper
6. Fix typos and make other minor writing improvements

We believe the reviews have definitely helped us improve our submission. We are running as many experiments as permitted by time and computation constraints and we hope the above mentioned changes will help readers understand the paper better.

---

### Decision · Program_Chairs · 2021-09-27

**Decision:**

Accept (Poster)

**Comment:**

The author response addressed most of the concerns of the reviewers and all reviewers recommend to accept the paper after the author response and discussion.

I recommend accept with the expectation that the authors will revise the paper for the camera ready, addressing the reviewer's concerns according to the author response, including, but not limited to the following aspects:
1. Add the mean and standard error to Table 1, 2 and Figure 4, 6
2. Add results on Pascal VOC
3. Add an extra ablation study on augmentations and batch sizes
4. Add ablation study for Mihai et al. and discuss it in the related work
5. Move the discussion on sentence length from the appendix to the main paper
6. Fix typos and make other minor writing improvements